# The Repurposing of Cellular Proteins during Enterovirus A71 Infection

**DOI:** 10.3390/v16010075

**Published:** 2023-12-31

**Authors:** Sudeshi M. Abedeera, Jesse Davila-Calderon, Christina Haddad, Barrington Henry, Josephine King, Srinivasa Penumutchu, Blanton S. Tolbert

**Affiliations:** 1Department of Biochemistry and Biophysics, Perelman School of Medicine, University of Pennsylvania, Philadelphia, PA 19104, USA; sudeshi.abedeera@pennmedicine.upenn.edu (S.M.A.); barrington.henry@pennmedicine.upenn.edu (B.H.); srinivasa.penumutchu@pennmedicine.upenn.edu (S.P.); 2Department of Chemistry, Case Western Reserve University, Cleveland, OH 44106, USA; jxd519@case.edu (J.D.-C.); christinahaddad621@gmail.com (C.H.); jak362@case.edu (J.K.); 3Howard Hughes Medical Institute, Chevy Chase, MD 20815, USA

**Keywords:** enterovirus A71 (EV-A71), ITAF, IRES, 5′-UTR

## Abstract

Viruses pose a great threat to people’s lives. Enterovirus A71 (EV-A71) infects children and infants all over the world with no FDA-approved treatment to date. Understanding the basic mechanisms of viral processes aids in selecting more efficient drug targets and designing more effective antivirals to thwart this virus. The 5′-untranslated region (5′-UTR) of the viral RNA genome is composed of a cloverleaf structure and an internal ribosome entry site (IRES). Cellular proteins that bind to the cloverleaf structure regulate viral RNA synthesis, while those that bind to the IRES also known as IRES trans-acting factors (ITAFs) regulate viral translation. In this review, we survey the cellular proteins currently known to bind the 5′-UTR and influence viral gene expression with emphasis on comparing proteins’ functions and localizations pre- and post-(EV-A71) infection. A comprehensive understanding of how the host cell’s machinery is hijacked and reprogrammed by the virus to facilitate its replication is crucial for developing effective antivirals.

## 1. Introduction

Viral infections can cause life-threatening illness. Enterovirus A71 (EV-A71), the causative agent of hand, foot, and mouth disease (HFMD), has resulted in high morbidity and in some cases mortality rates in infants and children under five years old [1]. Mild symptoms of HFMD include rashes on the hands, feet, and mouth, fever, and painful blister-like sores. In addition, infection with EV-A71 has been known to cause more severe complications such as brainstem encephalitis, cardiopulmonary failure, acute flaccid paralysis, and death in severe cases [1,2].

According to the World Health Organization (WHO), EV-A71 was first isolated in 1969 in California, USA; however, some studies showed that the virus could have emerged in 1963 when a worldwide epidemic occurred in the late twentieth century [3]. Outbreaks of EV-A71 have occurred primarily in East Asian countries as well as in Europe and America [4]. In China, there were approximately 7.2 million infections and 2 thousand deaths between 2008 and 2012 [5]. Waves of infections occurred in the Netherlands, France, and Germany from 2007–2013 [5]. Most recent incidences of EV-A71 infection include several cases in Spain in 2016 and around 13,000 cases in Gansu Province, China in 2018 [6,7]. Also in 2018, the Children’s Hospital of Colorado reported a 2.75 percent increase in confirmed cases as compared to the average from the past 5 years (2013–2017) [8]. In the same year, Vietnam reported 53,000 clinical cases of HFMD and 6 deaths [9]. Approximately 2500 Taiwanese individuals were infected with EV-A71 in 2020 and needed emergency or outpatient visits to the hospital [2]. Notably, the National Institute of Allergy and Infectious Diseases (NIAD) listed EV-A71 as an emerging infectious disease [10]. Due to the severity of EV-A71 infection and its life-threatening complications to children, researchers and scientists have been working on ways to combat this virus. To date, there is no antiviral treatment for EV-A71 infection; however, progress has been made in the development of vaccines. There are three inactivated, whole EV-A71 vaccines available in China (efficacy was at least 90% as confirmed in phase III clinical trials). Other types of vaccines are still in the development process [11,12].

EV-A71 belongs to the Picornaviridae family, genus Enterovirus, and species Enterovirus A [13]. It is a non-enveloped RNA virus with a positive sense, single-stranded RNA [14]. The viral genome of EV-A71 is around 7400 nucleotides long and composed of a 5′-untranslated region (5′-UTR), an open reading frame (ORF) that codes for viral polyprotein, and a 3′-UTR [14]. The 5′-UTR of the genome folds into a stable secondary structure that has six stem-loops (SLI-VI), as determined by structure prediction software. The SLI is also known as the cloverleaf structure while SLII-VI forms a type I internal ribosome entry site (IRES) (Figure 1b and Figure 2) [4,15,16]. The 5′-UTR region is a control hub for EV-A71 genome replication and translation.

Upon infection, EV-A71 causes rapid inhibition of the cap-dependent translation machinery of the host cell and utilizes host translation machinery for cap-independent translation of viral proteins mediated by its IRES element within the 5′-UTR (Figure 1a) [17]. Once the viral particle binds to the host cell receptor and the viral genome is released into the cytoplasm, translation results in a single polyprotein encoded by the ORF. This resultant polyprotein is cleaved by cellular and/or viral proteases, mainly 3C protease (3C^pro^) to generate a total of 11 viral proteins VP4, VP2, VP3, VP1, 2A, 2B, 2C, 3A, 3B, 3C, and 3D. As shown in the case of several enteroviruses that are related to EV-A71 such as polioviruses (PVs), coxsackieviruses (CVs), and rhinoviruses (RVs), 3C^pro^ along with 2A^pro^ also cleaves host translation initiation factors (eIFs): eIF4G and eIF5B, and PABP, resulting in the shutdown of host cell’s cap-dependent translation [17,18,19,20]. Inducing stress granule formation by 2A^pro^ and deactivation of eIF2α in response to the endoplasmic reticulum (ER) stress induced by EV-A71 infection further contributes to the shutdown of host translation [17]. It has been reported that PVs use a fragment of eIF5B resulting from 3C^pro^ cleavage as a substitute for eIF2α in order to continue viral RNA translation [21]. In addition, 3C^pro^ cleaves many other cellular factors to evade cellular responses against viral infections and also to induce cell apoptosis [22,23,24,25]. However, the information on the cleavage of diverse host factors specifically upon infection by EV-A71 awaits discovery.

For IRES-mediated translation to take place, 40S ribosomal subunit (43S ribosomal preinitiation complex) must be recruited on to the IRES elements mediated by eIFs [26], while auxiliary RNA-binding proteins (RBPs), which are known as IRES trans-acting factors (ITAFs), bind and differentially stabilize IRES elements to positively or negatively regulate the translation. Initiation of IRES-mediated translation occurs using only a subset of eIFs (as certain eIFs are degraded in order to inhibit host cap-dependent translation. The binding of ITAFs at specific regions of IRES can act as RNA chaperones and modulate IRES structure to make translation initiation feasible with the available subset of eIFs [27,28].

Picornaviruses have 5 types of IRESs (Type 1–5) and these IRESs are categorized based on the mechanism of recruiting 40S subunit to the start codon, RNA secondary structure, and ITAF requirements. Comparisons between the different IRES secondary structures and host factors required to recruit the ribosome have been reviewed elsewhere [27,29,30,31]. EV-A71, along with PV, human rhinovirus type 2 (HRV-2), bovine enterovirus (BEV), and coxsackievirus type B3 (CVB3), contains a Type 1 IRES structure. Type I IRES-containing viruses utilize a “land and scan” mechanism for start codon recognition by the ribosome, where ribosome is recruited at a landing site located at or immediately upstream of IRES SLVI followed by the translation initiation at the landing site or at a downstream site ribosome recognizes as it scans through the linker region between IRES and principal ORF [26,32,33,34]. The sequence and secondary structure of Type 1 IRES of a given virus is unique while it can have shared structural similarities with other Type 1 IRESs from different viruses. Hence, the host ITAF requirements for EV-A71 IRES differ and are unique from those of other related viruses containing Type 1 IRES elements. Exceptions can exist as in the case of PCBP2 which is implicated in the translation initiation of PV, EV-A71, and BEV [26].

Most ITAFs that are redistributed to the cytoplasm in response to viral infection and cellular stress are either nuclear or cycle between the nucleus and cytoplasm at varying degrees (Figure 1). This cytoplasmic redistribution of ITAFs is achieved mainly via 2A^pro^- and 3C^pro^-mediated or caspase-mediated pathways that target the nuclear pore complex to perturb nucleocytoplasmic trafficking [35]. Nucleocytoplasmic transport of proteins and RNA occurs through the nuclear pore complexes (NPC). These complexes are made of stationary and soluble components. Stationary components are different nucleoporins (Nups) where phenylalanine-glycine (FG)-Nups that contain intrinsically disordered FG-repeat domains are actively involved in the nucleocytoplasmic transport via interactions with soluble components of NPC, mainly nuclear transport receptors (karyopherins; including importins and exportins). Proteins that contain a nuclear localization signal (NLS) or nuclear export signal (NES) will be recognized by importins or exportins, respectively, to be imported into or exported out of the nucleus, respectively [35].

One of the main strategies of enteroviruses for the perturbation of nucleocytoplasmic trafficking is via the cleavage of specific components of NPC mainly FG-Nups mediated by 2A^pro^ while 3C^pro^ contributes to a lesser extent [35]. EV-A71 is also found to cleave karyopherin subunit α1 (KPNA1), a soluble component of NPC, in a 2A^pro^- and 3C^pro^-independent but caspase-dependent manner [36]. While the nuclear proteins are re-localized in the cytoplasm via perturbation of NPC, enterovirus proteases can also cleave NLS of proteins to prevent their import into the nucleus as observed in the case of 3C^pro^-mediated cleavage of the NLS of La autoantigen upon PV infection [37]. It is noteworthy that the perturbation of protein and RNA trafficking also results in cytoplasmic retention of transcription factors and blocks the mRNA export into the cytoplasm, thereby preventing host antiviral responses (e.g., interferon signaling) and making ribosomes available for efficient translation of viral RNA [35].

The majority of the ITAFs that have been discovered so far are positive regulators of EV-A71 translation. Some of the ITAFs also regulate viral replication via binding to the cloverleaf structure (SLI) of the 5′-UTR. Given the fact that the cellular localization and functional roles of most of these ITAFs have been changed upon EV-A71 infection, it is clear that EV-A71 naturally reprograms the cellular environment to achieve optimal viral replication. Here, we review up-to-date information on ITAFs of EV-A71 that have been identified so far (Table 1) and how they are repurposed during EV-A71 replication, which will aid in the process of developing more effective antiviral therapies against EV-A71 infection.

## 2. Cellular IRES Trans-Acting Factors (ITAFs) That Regulate EV-A71 Translation

### 2.1. hnRNP A1

Heterogeneous nuclear ribonucleoproteins (hnRNPs) are a family of diverse cellular proteins that play important roles in most aspects of RNA metabolism [38]. The quintessential member of this family of proteins is the abundantly expressed hnRNP A1. HnRNP A1 is primarily involved in the regulation of alternative splicing [39]; however, it is also known to modulate transcription, translation, stability, and nucleo-cytoplasmic export of mRNAs [40,41,42,43]. In addition to its regulatory functions in mRNA metabolism, hnRNP A1 is involved in microRNA (miRNA) processing, telomere maintenance, and modulation of transcription factor activity [40,44,45]. HnRNP A1 localizes to the nucleus but has the ability to transport mature mRNAs to the cytoplasm under stress-induced conditions [40,46,47]. HnRNP A1 achieves its varied functions in part due to its ubiquity and preference to bind short degenerate sequences, consisting of a 5′-YAG-3′ motif [48]. The original crystal structure of an hnRNP A1-RNA complex showed that its UP1 domain interacts specifically with the AG dinucleotide sequence via a “nucleobase pocket” formed by the β-sheet surface of RRM1 and the inter-RRM linker (Figure 3d) [49]. Other structures have shown that hnRNP A1 specifically binds 5′-YAG-3′ containing RNA sequences using various binding modes, revealing that it achieves cognate RNA recognition using context-dependent mechanisms [48,50,51].

Upon EV-A71 infection, hnRNP A1 re-localizes to the cytoplasm where it binds specifically to the SLII and SLVI domains of the viral IRES [15,52,53,54,55]. Individual knockdowns of hnRNP A1 and its homolog hnRNP A2 showed no significant effect on IRES activity, viral RNA synthesis, and viral 3C protease (3C^pro^). Interestingly, however, simultaneous knockdown of both hnRNP A1 and A2 resulted in a significant attenuation of IRES activity and reduction of viral titers by 4 log_10_ units, indicating that these two proteins are functionally interchangeable [52]. The SLII-hnRNP A1 interaction has been thoroughly characterized biophysically and functionally. HnRNP A1 binds to a phylogenetically conserved bulge loop to change the structure of SLII, which in turn stimulates IRES activity. Mutations or deletions to the bulge loop abrogate hnRNP A1 binding, significantly attenuate IRES translation and inhibit EV-A1 replication by ~5 log_10_ units [15,53,56]. Thus, the recruitment of hnRNP A1 to the EV-A71 5′-UTR is essential for efficient viral replication.

### 2.2. AUF1 (hnRNP D)

AU-rich element RNA-binding protein 1 (AUF1) also known as hnRNP D is found in 4 isoforms, p37, p40, p42, and p45, which are generated by alternative splicing and numbered based on their molecular weight [57]. HnRNP D is primarily responsible for mRNA decay via binding to AU-rich elements (AREs) in order to regulate mRNA turnover [57,58]. AREs are destabilizing elements found in most mRNA [59]. Additionally, AUF1 is involved in activating telomerase expression, repressing senescence, and maintaining normal aging [60]. AUF1 is also known to associate with several protein complexes [61,62,63].

Upon EV-A71 cell infection, AUF1 is transported from the nucleus to the cytoplasm and it associates with the EV-A71 5′-UTR [60]. AUF1 specifically binds the SLII of the IRES region [60,64]. The knockdown of AUF1 showed an increase in IRES activity, production of viral 3C^pro^, and viral titers with no effect on viral RNA synthesis or cap-dependent translation [43]. Hence, AUF1 is a negative regulator of IRES-dependent translation and viral replication.

Both AUF1 and hnRNP A1 bind to the same bulge region within the SLII domain. HnRNP A1 binding enhances IRES activity whereas AUF1 binding inhibits IRES activity in a putative mechanism that tunes the levels of IRES-dependent translation to meet the replication needs of EV-A71 [64]. A competitive binding assay between hnRNP A1 and AUF1 demonstrated that the levels of SLII-bound hnRNP A1 reduced upon increasing levels of AUF1 [65]. However, the combined knockdowns of hnRNP A1/A2 and AUF1 had no effect on IRES activity showing that IRES has an intrinsic function that can be tuned by these proteins as suggested by Lin et al. [65]. In addition to EV-A71, AUF1 has been reported to negatively regulate the replication of three related picornaviruses; PV, HRV, and CV in mammalian cells [65,66].

Small virus-derived RNAs (at least vsRNA1-4) are generated in EV-A71 infected cells due to the cleavage of EV-A71 5′-UTR by Dicer. Out of these vsRNAs, vsRNA1 is found to bind the SLII of the IRES to reduce IRES-mediated translation. It has been discovered that vsRNA1 promotes the association of AUF1 with SLII as well as the association of 2 other positive ITAFs, Ago2 and HuR, which will be discussed in detail later in this review. It is possible that vsRNA1 controls EV-A71 translation and replication via its ability to modulate selected ITAF-IRES interactions at SLII [17,67].

### 2.3. hnRNP K

HnRNP K is a versatile protein with functional roles in the nucleus as well as the cytoplasm [58]. It plays diverse functions in mRNA metabolism including regulation of transcription (via binding to CT-rich promoter regions), alternative splicing, mRNA silencing during cell differentiation (via interaction with GSK3β), mRNA stability (during cellular stress conditions), and translation [68,69,70,71,72]. HnRNP K interacts with RNA via recognizing CU-rich RNA patches while it can also interact with proteins using its Lysine (K)-rich domains [73].

Upon EV-A71 infection, hnRNP K that was localized in the nucleus will be enriched in the cytoplasm [54]. Lin et al. showed that hnRNP K binds to the 5′-UTR, specifically at the regions SL I-II and SLIV to regulate EV-A71 replication. The knockdown of hnRNP K in EV-A71 infected cells reduced viral replication and delayed the synthesis of positive and negative RNA strands [54]. These results indicate that hnRNP K positively regulates EV-A71 replication. The results by Lin et al. [54] do not show that hnRNP K enhances IRES-mediated translation even though it binds at the SLIV domain. However, the interaction of hnRNP K at SLIV may have a stabilization effect on EV-A71 that will promote viral RNA synthesis.

### 2.4. PCBP1 (hnRNP E1)

Poly(C)-binding protein 1 (PCBP1), also known as hnRNP E1, is a host protein located primarily in the nucleus and can be shuttled between the nucleus and cytoplasm. It has three K-homology (KH) domains that mediate RNA binding [74,75]. PCBP1 binds to CU-rich regions within the 3′-UTR of mRNA to increase its stability [76]. PCBP1 also has roles in modulating alternative splicing, mRNA silencing, transcription, and translation [77,78,79,80]. It is known that PCBP1 interacts with the 5′-UTR of PV RNA and facilitates viral RNA replication [75]. However, the mechanisms by which PCBP1 facilitates viral replication and translation of many viruses are not known.

Upon EV-A71 infection, PCBP1 primarily localizes in the cytoplasm and co-localizes with the EV-A71 RNA in the ER-derived membrane. PCBP1 specifically binds to the 5′-UTR at SLI and IV using its KH1 domain. Knockdown of PCBP1 decreased the level of VP1 viral protein while the overexpression of PCBP1 increased the VP1 production. Further, the knockdown of PCBP1 resulted in reduced EV-A71 viral titers while the overexpression of PCBP1 increased the titers. Hence, PCBP1 binds the EV-A71 5′-UTR and positively regulates viral protein expression and virus production [75].

### 2.5. PTB (hnRNP I)

The polypyrimidine tract-binding protein 1 (PTB or PTBP1), also known as hnRNP I, is an RNA-binding protein where it interacts with polypyrimidine stretches (CU repeats or CU-rich elements) on RNA mediated by four RNA recognition motifs (RRMs) (Figure 3e) [58,81]. Even though PTB is mostly known for its role in the regulation of alternative splicing of pre-mRNAs [82,83,84], it is also involved in various other cellular processes including mRNA stabilization, regulation of mRNA translation, and miRNA-mediated regulation of gene expression [85,86,87].

Upon cell infection, PTB is translocated from the nucleus to the cytoplasm to participate in viral processes. A study by Xi et al. discovered that PTB specifically binds to the SL VI (stem-loop VI and linker region, nt 564–742) of IRES using RRM1-2. Their results further demonstrate that PTB is a positive ITAF that enhances the EV-A71 IRES-mediated initiation of translation, viral protein expression, and virus production [88].

### 2.6. FBP2

The far upstream element binding protein 2 (FBP2 or FUBP2), which is also known as KH-type splicing regulatory protein (KHSRP or KSRP), is a host protein that plays diverse roles including regulation of transcription, pre-mRNA splicing, and mRNA editing [89,90,91]. FBP2 also regulates mRNA degradation via binding to AREs at 3′-ends as well as the maturation of miRNA precursors (Figure 3c) [92,93,94]. FBP2 can shuttle between the nucleus and the cytoplasm, and its localization can vary in a tissue- and context-dependent manner [93].

An extensive study by Lin et al. identified FBP2 as a cellular ITAF that binds to the 5′-UTR of EV-A71, in vitro and in vivo [55]. A pull-down assay suggests that FBP2 interacts with the regions SLI-II (nt 1–167), SLII-III (nt 91–228), and SLVI and spacer region (nt 566–745) (stem-loop VI and spacer region) in EV-A71 5′-UTR via a region that involves at least KH2 and KH4 domains. They further showed that FBP2 is relocalized from the nucleus to the cytoplasm during EV-A71 infection. Knockdown of FBP2 resulted in an increase in viral protein synthesis while FBP2 overexpression resulted in a decrease in viral protein synthesis. Further, FBP2 was shown to negatively regulate IRES-dependent translation to confirm the role of FBP2 as a negative regulator (negative ITAF) of IRES function. The competitive binding assay between PTB and FBP2 suggests that FBP2 acts as a negative ITAF via its ability to bind IRES competitively with positive ITAFs, such as PTB [55].

#### N- and C-Terminus Cleaved FBP2

Upon investigating FBP2, truncated products of the protein were detected in EV-A71-infected cells. The nonstructural proteins of EV-A71: 2A, 2B, 2C, 2BC, 3A, 3AB, 3C, and 3D are not responsible for the cleavage FBP2. However, the truncation of FBP2 is a result of viral replication that was found to occur through various virus-induced mechanisms involving caspase activation, proteasomes, and autophagy [95]. Five truncated products were detected, two of which, FBP2_1–503_ and FBP2_190–711_, bind to the 5′-UTR. Like full-length FBP2, FBP2_190–711_ negatively regulates viral translation, while FBP2_1–503_ is a positive regulator of IRES-driven translation and viral protein synthesis [95]. Hence, EV-A71 infection cleaves FBP2 at the C-terminus to reverse its function from a negative to a positive regulator of viral translation while retaining the ability of FBP2 to bind to the 5′-UTR [95].

### 2.7. FBP1

The far upstream element binding protein 1 (FBP1 or FUBP1), which is highly homologous to FBP2, can bind to RNA or ssDNA [96]. FBP1 activates the transcription of a proto-oncogene *c-myc* (cellular myelocytomatosis oncogene) that mediates cell growth by binding to the far upstream element (FUSE) upstream of the *c-myc* promoter [97]. It also inhibits the translation of nucleophosmin by binding to the 3′-UTR of its mRNA [98]. Additionally, FBP1 plays a role in post-transcriptional regulation of a growth-associated protein 43 (GAP43) in neural development. It promotes GAP43 mRNA degradation by binding to a pyrimidine-rich region at the 3′-UTR of the transcript [96]. FBP1 is known to interact with the UTRs Hepatitis C virus (HCV) and Japanese encephalitis virus (JEV) RNA genomes to positively and negatively regulate viral replication, respectively [99,100].

A study by Huang et al. identified FBP1 as a positive ITAF for EV-A71 replication [101]. Similar to FBP2, FBP1 naturally resides in the nucleus. However, upon EV-A71 infection, FBP1 is redistributed in the cytoplasm, where most of the steps in viral replication take place [101]. In vitro and in vivo studies show that FBP1 binds to the EV-A71 5′-UTR specifically at the linker region downstream of the IRES (nt 686–714). FBP1 contains four KH-type RNA binding domains (KH1-4) flanked by N- and C-terminal domains. However, KH3-4 is enough for FBP1 to bind the EV-A71 5′-UTR. FBP1 has demonstrated that it can positively regulate the IRES-dependent translation of EV71 via binding the 5′-UTR. FBP1 and FBP2 both bind to the linker region downstream of IRES but have opposing roles in IRES-mediated translation by acting as a positive and a negative regulator, respectively. An in vitro competition binding assay between FBP1 and FBP2 revealed that FBP1 outcompeted FBP2 in binding to the IRES linker region. This result suggests that FBP1 may act as a positive ITAF by preventing the negative ITAF, FBP2 binding to the IRES linker region [101].

#### C-Terminus Cleaved FBP1

EV-A71 induces the cleavage of FBP1 during the middle stages of infection. The viral protease 2A (2A^pro^) is responsible for the truncation of FBP1 into its primary cleavage product, FBP1_1-371_. However, unlike in the case of FBP2, the FBP1 cleavage is not due to proteasome, lysosome, or caspase activity [102]. Similar to intact FBP1, FBP1_1-371_ binds to the linker region of the 5′-UTR but at a different nucleotide sequence (nt 656–674). The non-competitive and simultaneous binding of FBP1 and FBP1_1-371_ additively enhance IRES-mediated translation as well as EV-A71 viral yield [102].

### 2.8. Ago2

The Argonaute 2 (Ago2) protein is implicated in transcriptional and post-transcriptional gene silencing [103]. Ago2 is a component of the RNA-induced silencing complex (RISC), where Ago2 binds to a guide RNA, such as a microRNA or a short interfering RNA (Figure 3a) [103,104]. Silencing occurs through binding of this guide RNA to a complementary strand on the target mRNA [103,104]. This will either cause an endonucleolytic cleavage of the mRNA by Ago2 or the inhibition of translation [103,104,105]. The silencing pathway chosen is based on the degree of complementarity between the guide RNA and its target mRNA [103]. In addition, Ago 2 was found to upregulate translation via binding to AU-rich elements at the 3’-UTR [106].

Ago2 is cellularly localized in the nucleus and the cytoplasm [105]. However, Ago2’s localization upon EV-A71 infection has not been reported. Ago2 was identified as an ITAF that specifically binds at the SLII of EV-A71 5′-UTR. The knockdown of Ago2 resulted in reduced IRES activity, demonstrating that Ago2 is a positive ITAF. Further, Ago2 knockdown resulted in a reduction of 3C^pro^ expression (25–35%) and viral yields (10^3^ fold) [67].

### 2.9. HuR

Human antigen R (HuR), also known as ELAV-like RNA-binding protein 1 (ELAVL1), binds various mRNAs via recognition of AU-rich elements to regulate their stability and translation (Figure 3f) [107,108]. HuR is predominantly located in the nucleus but shuttles between the nucleus and cytoplasm [109]. The N-terminal RRM1 and RRM2 of HuR mediates its interactions with poly-U or AU-rich elements while RRM3 mediates its interactions with the poly-A tail of target mRNAs [109,110]. HuR has also been shown to positively regulate the expression of Ago2 [111].

Upon EV-A71 infection, HuR previously localized primarily in the nucleus is translocated into the cytoplasm [112]. HuR was identified as an ITAF that specifically binds at the SLII of EV-A71 5′-UTR. The knockdown of HuR resulted in reduced IRES activity, demonstrating that HuR is a positive ITAF. Further, HuR knockdown resulted in a reduction of 3C^pro^ expression (25–35%) and viral yields (10^3^ fold) [67].

#### Additive Effects of Ago2 and HuR

The simultaneous knockdown of Ago2 and HuR did not show an additive effect on the reduction of IRES activity, while the effect was additive on the reduction of 3C^pro^ expression (2.5 fold) and viral yields (10^6^ fold). However, neither independent nor simultaneous knockdown of Ago2 and HuR had an effect on viral RNA synthesis. These results suggest that Ago2 and HuR influence EV-A71 replication by acting as a positive regulator of IRES-dependent translation [67].

As described here, there are several ITAFs that bind SLII of EV-A71 5′-UTR, including Ago2, HuR, AUF1, and hnRNP A1/A2 [52,65,67]. Competitive binding studies were carried out to study the dependency of binding of one protein to another. Based on the results from knockdown assays, AUF1, Ago2, or HuR did not depend on one another to bind to SLII. However, an increase in vsRNA1 caused an increase in the binding of AUF1, Ago2, and HuR to SLII, while the binding of hnRNP A1/A2 was unaffected. Given that AUF1 is a negative regulator and while HuR and Ago2 are positive regulators of IRES activity, it is speculated that vsRNA1 binding at SLII fine-tunes the binding of AUF1, HuR, and Ago2 at SLII to regulate viral IRES-mediated translation [67]. Thus, it can be deduced that SLII acts as a coordination hub to recruit different ITAFs which in turn modulate viral protein synthesis, and that vsRNA1 further sculpts the SLII-ITAF interactions to regulate viral gene expression.

### 2.10. MOV10

The Moloney Leukemia Virus 10 (MOV10) is a cellular RNA helicase protein found in the cytoplasm [113,114]. It colocalizes with Ago2 in the RISC complex to mediate microRNA-guided mRNA cleavage [115]. MOV10 mediates mRNA degradation of thousands of mRNAs by initially binding to the 3′-UTR and then translocating along the 3′-UTR to unfold the structure and disassemble proteins prior to mRNA degradation [114]. The C-terminus of MOV10 consists of seven highly conserved helicase motifs that account for its RNA binding activity, while its N-terminal CH-domain mediates protein–protein interactions [116].

A study by Wang et al. has demonstrated that MOV10 positively regulates EV-A71 replication [113]. The knock down of MOV10 drastically reduced the levels of viral protein as well as positive-strand RNA. MOV10 was found to interact with SLI, the cloverleaf-like structure, and the IRES of E-AV71 5′-UTR to facilitate viral RNA replication and IRES-dependent translation, respectively. Further, it was shown that MOV10 brings out the positive regulation of EV-A71 replication through its C-terminus (the site of RNA helicase activity), while the N-terminus possesses a potentially inhibitory effect on viral production by inhibiting viral translation via an unknown mechanism. Upon EV-A71 infection, MOV10 formed distinct perinuclear aggregates and co-localized with processing bodies (P-bodies). In addition to its interaction with Ago2, MOV10 was shown to interact with HuR, another positive ITAF that interacts with the EV-A71 5′-UTR [113].

### 2.11. SIRT1

Silent mating-type information regulation 2 homolog 1 (SIRT1) is a member of the sirtuin family. It is a NAD^+^-dependent deacetylase and is involved in a broad range of physiological functions including control of gene expression, metabolism, and aging [117]. SIRT1 has been implicated in obesity-associated metabolic diseases, cancer, aging, cellular senescence, neurodegeneration, and inflammatory signaling in response to environmental stress [117,118,119]. SIRT1 has been known to positively regulate human immunodeficiency virus 1 (HIV-1) transcription and hepatitis B virus (HBV) replication through the deacetylation of Tat and transcription factor AP-1, respectively [120,121].

SIRT1 is localized in the nucleus; however, upon EV-A71 infection, the protein is translocated from the nucleus to the cytoplasm [122]. According to Han et al., SIRT1 interacts with the SLI (cloverleaf structure) and the SLs II, III, and V within IRES of EV-A71 5′-UTR to repress viral RNA replication and IRES-mediated translation, respectively. At the same time, SIRT1 binds viral 3D^pol^ protein and results in the repression of viral genome replication [122].

### 2.12. Sam68

The 68 kDa Src-associated protein in mitosis (Sam68), also known as the KH domain-containing, RNA-binding, signal transduction-associated 1 (KHDRBS1) protein, is implicated in many processes including cell cycle and signaling, cell growth, alternative splicing, pre-mRNA splicing, and trafficking [123,124,125]. Sam68 belongs to the signal transduction and activation of RNA (STAR) protein family as well as the hnRNP K homology (KH) domain family of RNA-binding proteins [126]. It contains one KH domain that interacts with RNA and several proline-rich sequences that facilitate protein–protein interactions with SH3- and WW domain-containing proteins [127]. The KH domain of Sam68 recognizes and binds RNA using U(U/A)AA direct repeat motifs [128,129].

Sam68 is an ITAF that resides in the nucleus and gets relocated to the cytoplasm upon EV-A71 infection. Similar redistribution of Sam from the nucleus to the cytoplasm has also been reported in cases of FMDV and PV infections [130,131]. Sam68 specifically interacts with the SLIV and V of 5′-UTR of EV-A71 RNA using its hnRNP K homology (KH) domain. Sam68 was found to interact with other ITAFs including Poly(C)-binding protein 2 (PCBP2) and Poly(A)-binding protein (PABP) to facilitate viral replication. Sam68 is a positive regulator of IRES activity, viral protein expression, and viral titer. However, Sam68 does not have any regulatory effect on viral genome replication [132].

### 2.13. FUBP3

The far upstream element-binding protein 3 (FUBP3) is a single-stranded NA-binding protein that recognizes only one strand of the far upstream element (FUSE) [90]. The structure of FUBP3 consists of four regularly spaced K homology (KH) domains that recognize similar sequences in single-stranded DNA or RNA targets [133]. FUBP3 has roles in the regulation of transcription, splicing, and translation [134]. FUBP3 is known to bind the 3′-UTR of the Japanese encephalitis virus (JEV) to regulate RNA replication and promote subsequent viral translation and viral particle production [133].

A study by Tsai et al. on the inhibition of EV-A71 replication and internal ribosome entry site (IRES) activity by Kaempferol (a flavonoid) has discovered FUBP3 as a new ITAF that associates with EV71 5′-UTR to enhance the IRES-dependent translation [135]. Another study by Huang et al. has reported the interaction of FUBP3 with the 5′-UTR of EV-A71 to regulate its replication in differentiated neuronal cells [136]. Upon EV-A71 infection, FBP3 in the nucleus will be relocated into the cytoplasm [136].

### 2.14. GADD34

The growth arrest and DNA damage-inducible protein 34 (GADD34), also known as PPP1R15A, is a protein that is upregulated in response to various cell stress-inducing stimuli. GADD34 interacts with serine/threonine protein phosphatase 1 (PP1) to dephosphorylate eIF2α, thereby restarting protein synthesis for cells to recover from an integrated stress response (ISR) [137,138,139]. GADD34 is known to play an important role in regulating the interferon response of virus-induced innate immunity [140,141]. While there are many reports regarding the role of GADD34 in the inhibition of viral replication [140,141,142], GADD34 has been shown to promote the replication of infectious bronchitis virus (IBV) [143]. GADD34 attenuated HIV-1 replication via inhibition of viral protein expression in a mechanism mediated by 5′-UTR/TAR RNA, probably by modulating TAR RNA structure [144].

A study by Li et al. demonstrated that EV-A71 activates GADD34 via viral precursor protein 3CD to promote IRES-mediated viral translation. GADD34 is a short-lived protein and is highly expressed only under the conditions of cellular stress [145]. Upon EV-A71 infection, 3CD upregulates GADD34 translation via the upstream open reading frame (uORF) within the 5′-UTR of GADD34. GADD34 is unable to bind directly to EV-A71 5′-UTR. Hence, 3CD binds at the SLI (cloverleaf structure) and recruits GADD34 to the 5′-UTR of EVA-A71. Once bound at the 5′-UTR, GADD34 promotes the EV-A71 IRES activity through its PEST (P: proline, E: glutamate, S: serine, and T: threonine) repeats (1, 2, and 3). However, the dephosphorylation of eIF2α by GADD34 was unrelated to these observed effects in the upregulation of EV-A71 replication [146].

### 2.15. DDX3

DEAD-box protein 3 is a DEAD-box RNA helicase that regulates translation and is encoded by the X- and Y-linked paralogs *DDX3X* and *DDX3Y*. DDX3X is ubiquitously expressed and essential for viability while DDX3Y is male specific and shows lower and more variable expression in somatic tissues compared to DDX3X. However, the roles of DDX3X and DDX3Y in translation are functionally redundant [147]. DDX3 has a relaxed substrate specificity and is implicated in many cellular processes such as gene expression including transcription, splicing, mRNA export, translation, cell cycle control, regulation apoptosis, and innate immune signaling (Figure 3g) [148]. It has been known that DDX3 is a prime target for viral manipulation during Hepatitis C virus (HCV), Hepatitis B virus (HBV), Human Immunodeficiency Virus (HIV), and poxvirus infections as viral proteins interact with DDX3 to utilize its function for the process of viral replication [148]. It has been shown that DDX3X is specifically required for the translation initiation of transcripts that possess highly stable secondary structures within their 5′-UTR that resist the unwinding activity of eIF4A. DDX3X binds the 5′-UTR via its interactions with eIF4G and PABP, and works in corporation with eIF4A to destabilize the secondary structure and facilitate ribosome entry [149].

DDX3X shuttles between the nucleus and the cytoplasm and possesses RNA-dependent ATPase/helicase activity [148]. Su et al. demonstrated that DDX3X binds the 5′-UTR of EV-A71 and enhances IRES-dependent translation, partly mediated by viral 2A^pro^ and 3C^pro^ protease activity. DDX3X can bind efficiently to IRES + spacer, SLI-III, SLIV-VI, and SLVI + spacer regions in the EV-A71 5′-UTR, irrespective of EV71 infection. Their results strongly suggest that the truncated eIF4G (cleaved by viral 2A^pro^) binds specifically to SLV and recruits DDX3X to SLVI or a region nearby to locally unwind the secondary structure of SLVI, thereby facilitating ribosome entry and scanning. They also showed that DDX3X also enhances the IRES activity of coxsackievirus A16, Echovirus 9, EMCV, and HCV implicating DDX3X is a general cellular factor for the translation of these highly structured viral IRESs [150].

### 2.16. APOBEC3G

APOBEC3G (apolipoprotein B mRNA-editing enzyme, catalytic polypeptide-like 3G) or A3G is a member of the APOBEC superfamily. A3G is a cytidine deaminase that contains a conserved His-X-Glu and Cys-X-X-Cys Zn^2+^ coordination motif. A3G is an interferon-inducible cellular protein and plays an important role in defending against viral infections. It has been demonstrated to inhibit the infection of several viruses such as human immunodeficiency virus-1 (HIV-1), T-cell leukemia virus type 1 (HTLV-1), hepatitis B virus (HBV), and Hepatitis C virus (HCV) [151,152]. A3G can inhibit viral replication in a cytidine deaminase activity-dependent (C-terminal domain) manner, as well as in a deaminase activity-independent manner, where it is the N-terminal domain’s RNA binding ability that mediates incorporation of A3G into viral particles, thereby disrupting reverse transcription or genome encapsidation [153,154,155].

Li et al. demonstrated that A3G inhibits EV-A71 virus replication via competitive binding to the 5′-UTR of EV-A71 and inhibiting the 5′-UTR activity. A3G binding impaired the interaction between the 5′-UTR and the host protein poly(C)-binding protein 1 (PCBP1), an ITAF that enhances the viral RNA replication and IRES-mediated translation. A3G was shown to bind the SLI and II while PCBP1 is known to bind to SLI and IV of EV-A71 5′-UTR [75,156]. Hence, A3G binds at SLI and competitively inhibits the binding of PCBP1 at the same location, as suggested by the higher binding affinity of A3G compared to PCBP1 towards the 5′-UTR and the reduction of 5′UTR-PCBP1 interactions with increasing A3G expression levels. However, EV-A71 has developed mechanisms to overcome the suppression by A3G via its degradation through the autophagy–lysosome pathway mediated by viral protein 2C^pro^ [156].

### 2.17. Staufen1

Staufen1 is one of the two homologs of Staufen, a double-stranded RNA (dsRNA) and tubulin-binding protein [157]. Staufen1 contains four dsRNA-binding domains (RBDs) of which dsRBD2-4 are shown to bind dsRNA (Figure 3h) [158]. Staufen1 is known for its ability to regulate cellular mRNA translation, trafficking, and degradation via Staufen1-mediated RNP formation. Staufen1 enhances the translation efficiency via binding to the 5′-UTR while it can bind to the 3′-UTR to promote mRNA degradation [27,159]. A study by Chen et al. has demonstrated that Staufen1 facilitates both translation and replication of the EV-A71 genome. The RBD2-3 of Staufen1 was identified to interact with the 5′-UTR EV-A71 to enhance the IRES activity as well as the translation efficiency. Further, the binding of Staufen1 at the 5′-UTR increased the stability of viral RNA [160].

Staufen1 has been reported to play roles in the life cycles of other RNA viruses, including Hepatitis C virus (HCV), influenza A virus, and HIV-1. During the infection of HCV, Staufen1 demonstrated roles in the viral replication, translation, and trafficking of the HCV genome [161], while it facilitated the viral particle assembly of HIV-1 and Influenza A viruses [162,163].

### 2.18. hnRNP H and hnRNP F

The two hnRNP proteins, hnRNP F and hnRNP H, are closely related and fall under the hnRNP F/H subfamily. They are found to act as activators as well as repressors in regulating alternative splicing, depending on the context of the binding site [164]. HnRNP F/H possess three quasi-RNA recognition motifs (qRRMs) that preferably bind to poly(G)-rich sequences in the target exons and/or adjacent introns in order to regulate alternative splicing and 3′-end processing of numerous genes (Figure 3b) [165].

Both hnRNP H and F are located in the nucleus and are relocated into the cytoplasm upon EV-A71 infection. A study by Tsai et al. on the inhibition of EV-A71 by Kaempferol (a flavonoid) via impairing its replication and internal ribosome entry site (IRES) activity has discovered hnRNP F and hnNNP H as new ITAFs that associates with EV-A71 5′-UTR to enhance the IRES activity. Both hnRNP H and F were discovered initially as kaempferol-induced cellular factors associated with the 5′-UTR of the EV-A71 genome. However, they have carried out individual knockdown of endogenous HNRH1 and HNRPF proteins to show that it resulted in decreased EV-A71 IRES activity. Unfortunately, the specific binding sites of hnRNP H and F on the EV-A71 5′-UTR have not been mapped [135].

### 2.19. EGR1

Early growth response-1 (EGR1) is a C_2_H_2_-type zinc finger protein, and it is a transcription factor that activates many genes essential for growth, proliferation, or differentiation via binding to GC-rich recognition motifs. EGR1 expression is induced in response to various extracellular stimuli including growth factors, hormones, and neurotransmitters. EGR1 then couples extracellular signals to long-term cellular responses by altering the gene expression of its target genes [166]. EGR1 is found to activate microRNA-141 (miR-141) expression to suppress eukaryotic initiation factor 4E (eIF4E) production, resulting in the facilitation of EV-A71 replication via shutting off host protein synthesis [167].

A study by Song et al. has revealed that the first two zinc fingers of EGR1 bind directly to SLI and SLIV of the EV-A71 5′-UTR to regulate viral replication via enhancing both IRES-mediated translation and RNA replication. Further, this study demonstrated that EGR1 facilitates EV-A71 replication in a manner independent of miR-141 and eIF4E, via its interactions with the 5′-UTR [168].

### 2.20. TIA-1 and TIAR

T-cell intracellular antigen 1 (TIA-1) and TIA-1-related protein (TIAR) have been identified as nucleating components of mammalian stress granules (SGs) [169]. TIA proteins consist of three RNA-binding domains and a glutamine-rich carboxyl-terminal domain that enables aggregation to insoluble aggregates. TIA proteins have many important roles in the regulation of mRNA metabolism, especially during environmental stress conditions, including the regulation of splicing, mRNA stability, storage, and translation efficiency. The TIAR/TIA-1 proteins are known to selectively target specific mRNAs via binding to specific AREs located at the 3′ end of the target mRNAs [170]. TIA-1 and TIAR have been reported to be translocated from the nucleus to the cytoplasm to form SG-like granules as a result of PV and enterovirus (CVB3 or EV-A71) infections [171,172].

Wang et al. demonstrated that TIA1 and TIAR were recruited into SGs following EV-A71 infection and were localized to the sites of viral replication. Both TIA-1 and TIAR interact only with the SLI (nt 1–105) of EV-A71 5′-UTR and result in positive regulation of viral replication. The silencing of either TIA-1 or TIAR expression significantly reduced not only viral replication but also viral progeny production, via regulation of the level of viral RNA. Even though TIA-1 and TIAR bind to AREs, RNA pull-down assays proved that they did not bind to 5′-UTR or 3′-UTR of EV-A71 genome, except for SLI, making both TIA1 and TIAR distinct from many ITAFs that are shown to interact with multiple loops within 5′-UTR [173].

### 2.21. Additional Cellular Proteins

In addition to the proteins listed in Table 1, there are many additional cellular proteins that were discovered to associate with EV-A71 5′-UTR while their specific binding sites and roles in EV-A71 viral replication have not yet been determined. These proteins were identified by pull-down assays utilizing biotin-labeled 5′-UTR followed by MALDI-TOF MS characterization. Such additional cellular proteins that were discovered by individual studies carried out by Lin et al. and Xi et al. are as follows: TBP-associated factor TAFII 150 (TBP-AFII 150), N-ras upstream protein (unr), glycyl-tRNA synthetase (GARS), keratin, IGF-II mRNA-binding protein 1 (IMP-1), polypyrimidine tract-binding protein 2 (PTB-2), ErbB3-binding protein (EBP1), poly(rC)-binding protein 2 (PCBP2), pro alpha (I) collagen, Lrp protein, PTB-associated splicing factor (PSF), splicing factor, proline and glutamine rich (SFPQ), cationic trypsinogen, CGI-55 protein, elongation factor 1-gamma (EEF1G), 2-phosphopyruvate-hydratase alpha-enolase, collagen-binding protein 2, mutant beta-actin, and DNA-binding protein [54,55,88].

## 3. Discussion

Upon EV-A71 infection, the host cellular pathways will be repurposed to facilitate viral replication. Some of the main pathways that are perturbed include innate immunity, RNA metabolism, nuclear-cytoplasmic transport, stress granule formation, and autophagy. EV-A71 uses two main strategies to achieve the repurposing of cellular machinery. The first strategy is the redistribution of cellular proteins from their original locations into the cytoplasm where viral replication takes place. This will perturb their original cellular functions while being utilized solely in different steps of viral replication. The second strategy is the cleavage of key cellular proteins involved in above mentioned cellular pathways, mediated by viral proteases (mainly 3C^pro^ and 2A^pro^) or other virus-induced pathways (caspases, proteosome, and autophagy) (Figure 4) [17,27].

As the viral RNA genome is released into the cytoplasm upon EV-A71 infection, viral proteins are translated in an IRES-dependent manner while the viral genome is replicated via mechanisms mediated by RNA-dependent RNA polymerase (RdRp) 3D^pol^ [174]. The 5′-UTR of the viral RNA genome, which is composed of the cloverleaf structure (SLI) and the IRES (SLII-VI), functions as a control hub for viral RNA replication and translation via recruiting viral and cellular proteins (Figure 1). Cellular RNA-binding proteins that are recruited to the IRES and regulate viral translation are called ITAFs. The regulatory effect of these ITAFs on viral IRES-mediated translation can either be positive or negative [15,16]. Table 1 summarizes the ITAFs associated with the EV-A71 5′-UTR indicating their effect on viral translation and how they are redistributed in the cell upon EV-A71 infection.

It is noteworthy that hnRNP proteins are a key target of EV-A71, possibly due to the fact that hnRNP family proteins are central players in regulating various aspects of mRNA metabolism including alternative splicing, stability, transcription, and translation, while each of them is structurally and functionally diverse [58]. The hnRNP proteins are primarily localized in the nucleus and upon EV-A71 infection, they are relocated into the cytoplasm where viral replication takes place. Out of the seven hnRNP proteins listed in this review, hnRNP A1, hnRNP E1 (PCBP1), hnRNP I (PTB), hnRNP H, and hnRNP F act as positive ITAFs that bind EV-A71 5′-UTR to enhance IRES-mediated viral translation. The only hnRNP that acts as a negative ITAF is hnRNP D (AUF1). The hnRNP proteins that act as positive ITAFs also play simultaneous roles in other steps of EV-A71 replication such as viral genome replication.

In addition to the hnRNP proteins, FBP1, Ago2, HuR, MOV10, Sam68, FBP3, GADD34, DDX3, HSPA6, Staufen1, and EGR1 act as positive ITAFs, while only FBP2, SIRT1, and APOBEC3G act as negative ITAFs. However, EV-A71 has strategies to repurpose even the negative ITAFs into positive ITAFs. One such example is the cleavage of the C-terminus of FBP2 to convert the negative ITAF into a positive ITAF [102]. ITAFs can have different and unique cellular localizations that are related to their cellular function; however, they will be redistributed into the cytoplasm to bind the EV-A71 5′-UTR and enhance the IRES-mediated translation. Figure 1 visually represents the re-localization of cellular proteins before and after EV-A71 infection as well as their binding sites at the 5′-UTR of the EV-A71 RNA genome.

Apart from the relocation of cellular ITAFs into the cytoplasm, the levels of mRNA as well as the translation of positive ITAFs can also be enhanced upon EV-A71 infection, as observed in the case of GADD34 [146]. Further, EV-A71 viral proteins such as 2A^pro^ or 3C^pro^ play important roles in the process of repurposing cellular proteins to facilitate viral replication, mainly via their protease activity. Their activity is known to facilitate the redistribution of positive ITAFs from the nucleus to the cytoplasm via cleaving certain components of the nuclear pore complex [35]. Sometimes, their activity can truncate a cellular protein to give it a new feature that is beneficial for viral replication as observed in the case of eIF4G, where a truncated version of eIF4G (cleaved by viral 2A^pro^) binds specifically to SLV of IRES and recruits positive ITAF DDX3X facilitating ribosome entry and scanning [150,175]. In some cases, even positive ITAFs are cleaved by viral proteases as observed in the cleavage of FBP1 by 2A^pro^ to give rise to an additive enhancement of IRES activity [102] (Figure 4).

It is noteworthy that proteins such as poly(C)-binding protein 2 (PCBP2) and glycyl-tRNA synthetase (GARS) that are known to play important roles in IRES-mediated translation initiation by Type 1 IRES structures are not well studied on EV-A71. It has been shown that the translation initiation by PV, EV-A71, and BEV Type 1 IRESs necessitate PCBP2 at SLIV in addition to different eIFs [26,176,177,178]. GARS has been shown to bind SLV of the PV IRES adjacent to the binding site of the key initiation factor eIF4G, thereby enhancing the IRES activity [179]. In addition to PCBP2 and GARS, there are many other ITAFs such as La, SRp20, and upstream of N-Ras (unr) that have been implicated in the initiation of Type 1 IRES-mediated translation initiation during other enterovirus infections [180,181,182]. Most of these ITAFs have been discovered to associate with EV-A71 5′-UTR as indicated in Section 2.21: Additional Cellular Proteins, using pull-down followed by MS analysis, while their specific binding sites and roles in EV-A71 viral replication are yet to be determined.

The mechanism of fine-tuning the IRES activity to regulate EV-A71 viral replication is a very complex process involving the interplay between the positive and negative cellular ITAFs as well as the viral proteins. Several recent studies have expanded the repertoire of positive and negative ITAFs suggesting new targets for EV-A71 inhibition. Most importantly, studies that identify the interplay between positive and negative ITAFs that bind at the same site, not only help in understanding the means of IRES-mediated translation regulation but also highlight these sites as better candidates for drug targeting. We believe that a comprehensive and generalized understanding of the mechanism of how EV-A71 hijacks and repurposes host cellular proteins for its gain can guide the development of novel and efficacious approaches for drug targeting.

## 4. Future Perspectives

Considering the increased interest in discovering novel small molecule inhibitors of RNA viruses like EV-A71, it is important to implement robust strategies to identify biologically relevant targets for therapeutic intervention. The 5′-UTR of EV-A71 represents a promising target given its central importance in regulating the cellular stages of viral replication cycles. Moreover, the EV-A71 5′-UTR is under high selective pressure to maintain its overall structure because it associates with a collection of ITAFs to differentially control viral gene expression. The next generation of EV-A71 antivirals to be discovered should therefore function by selectively perturbing specific 5′-UTR-ITAF interactions to drive viral replication towards predictable outcomes (Figure 1 and Figure 2). Hence, small molecules that successfully perturb specific SLI-ITAF and IRES-ITAF interactions will function as novel antivirals against EV-A71 via inhibition of viral RNA replication and translation, respectively.

To date, SLII is the only RNA structure of the EV-A71 5′-UTR where specific binding sites of several of its ITAFs have been biophysically investigated to reveal the competitive interplay between hnRNP A1 and AUF1 for its bulge loop [53,65,183]. Further, this region has been successfully targeted by a small molecule antiviral DMA-135 (2-log and 5-log reduction of viral titers at 0.5 and 50 µM of DMA-135, respectively; IC_50_ of 7.54 ± 0.0024 μM) where its mechanism of action is to allosterically increase the binding affinity of AUF1 to SLII, thus shifting the SLII regulatory axis towards translation repression [64]. Notably, EV-A71 can evolve drug resistance to DMA-135 by changing nucleotides that form part of the AUF1 binding site, while not perturbing the binding activity of hnRNP A1 [56]. These collective observations suggest that other small molecules with the capacity to selectively perturb specific 5′-UTR-ITAF interactions await discovery.

To realize this outcome, we believe that it is prudent to better understand the structural-based mechanisms by which ITAFs assemble onto the EV-A71 5′-UTR, determine the extent to which subsets of ITAFs cooperate to enhance or suppress IRES-dependent translation and how they interact with eIFs, characterize the differential cell-type expression patterns of ITAFs upon viral infection, and perform comparative virological and structural studies of related enteroviruses that contain Type 1 IRES elements, such as EV-D68. Lastly, we must also remember that ITAFs and viral RNA elements are covalently modified so it should be a priority to understand the extent to which such modifications modulate EV-A71 gene expression at a molecular level [184]. We believe that the coordination of these studies will accelerate the discovery of the next generation of antiviral agents that target viral RNA complexes and produce a library of chemical biology reagents that can be deployed to selectively modulate conserved host–virus interactions with the potential to reveal new biological functions.

## Figures and Tables

**Figure 1 viruses-16-00075-f001:**
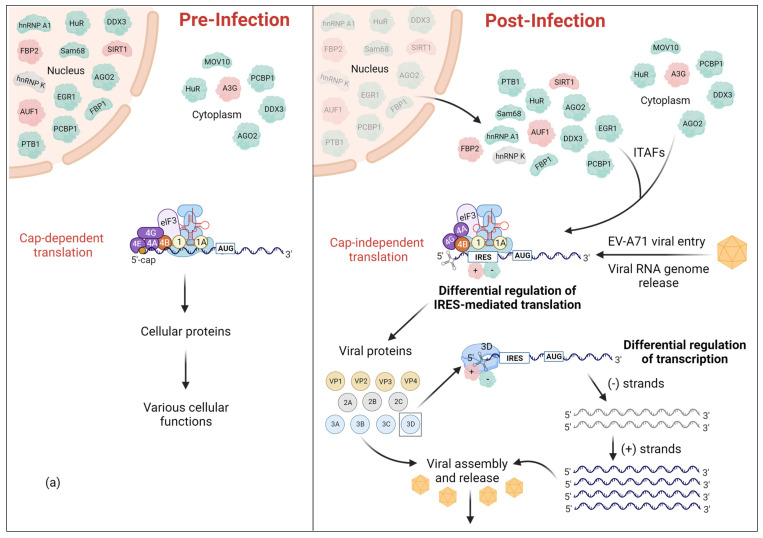
Reprogramming of the cellular environment upon EV-A71 infection. Following viral infection, several host RNA binding proteins, collectively referred to as ITAFs, change their sub-cellular localization. (**a**) These cellular ITAFs are primarily localized in the nucleus or cytoplasm while some of them can be shuttled between nucleus and cytoplasm based on their cellular functions. Regardless, upon EV-A71 infection, all ITAFs will be relocalized into the cytoplasm to regulate IRES-mediated translation. (**b**) The 5′-UTR of EV-A71 genome consists of the cloverleaf structure (SLI) and the IRES (SLII-VI). Cellular ITAFs bind to specific region(s) within IRES to either positively or negatively regulate IRES-mediated translation. Positive and negative ITAFs are indicated in green and red, respectively. The ITAFs, of which the effect on translation is not discovered yet, are indicated in grey. Some ITAFs can also bind the SLI region to regulate viral replication.

**Figure 2 viruses-16-00075-f002:**
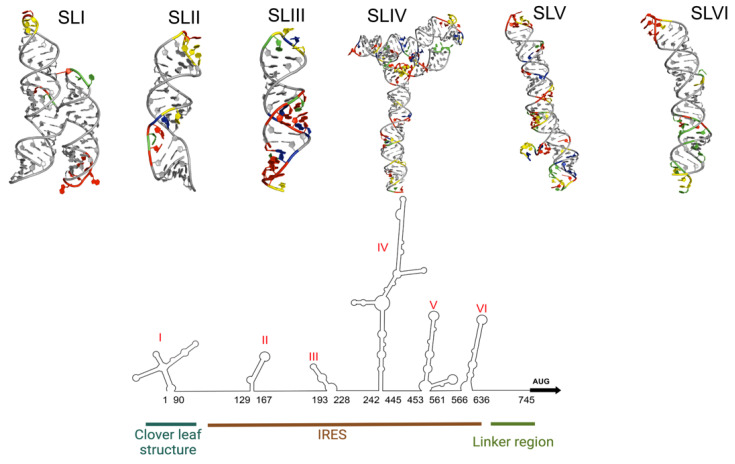
The EV-A71 5′-UTR contains phylogenetically conserved stem-loops (SLs) that adopt well-defined 3D structures. Depicted is a predicted secondary structure of the 5′-UTR of EV-A71, and 3D models of each individual stem-loop. The four nucleotides: adenosine (red), uridine (green), guanosine (blue), cytidine (yellow) are color coded in the regions that are predicted to be single stranded and most likely to interact with listed ITAFs. The base-paired nucleotides are shown in grey. 3D structures of SL I and II are published PDB structures while SLs III–VI were produced using the FAFFAR module of the ROSETTA software suit. The lowest energy structure for each stem-loop has been selected from a pool of 10,000 structures. The structures of SLI (8DP3) and SLII (5V17) were obtained from the PDB. Note, the structure of SLI of EV-A71 has not been published yet. Shown here is the SLI of the coxsackievirus B3 (CVB3) that was selected due to the high secondary structural homology between SLI regions of EV-A71 and CVB3.

**Figure 3 viruses-16-00075-f003:**
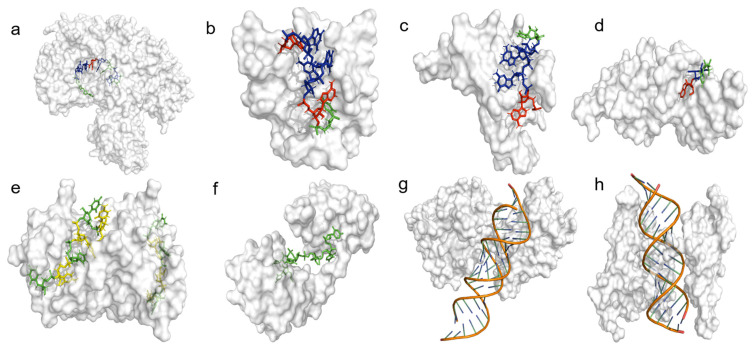
Cellular ITAFs bind with specificity to short and degenerate sequence motifs. Depicted are published PDB structures of cellular ITAFs bound to RNA. These structures have been experimentally determined by both X-ray and nuclear magnetic resonance (NMR) spectroscopy. RNA has been color coded: adenosine (red), uridine (green), guanosine (blue), cytidine (yellow). ITAFs that bind to double-stranded RNA have the RNA depicted as a cartoon structure. The identity and PDB ids of the depicted ITAFs are as follows: (**a**) Argonaute 2 (5KI6), (**b**) hnRNP F (2KFY), (**c**) Far -Upstream Binding Protein 2 (4B8T), (**d**) hnRNP A1 (4YOE), (**e**) Polypyrimidine tract-binding protein 1 (2N3O), (**f**) Human Antigen R (6G2K), (**g**) DEAD-Box Protein 3 (6O5F), (**h**) Staufen homolog 1 (6HTU). Note, the crystal structure of the hnRNP A1-RNA complex (4YOE) was solved with a 5′-AGU-3′ sequence bound to only its RRM1 domain, confirming that a single-strand AG dinucleotide is the minimal specificity motif.

**Figure 4 viruses-16-00075-f004:**
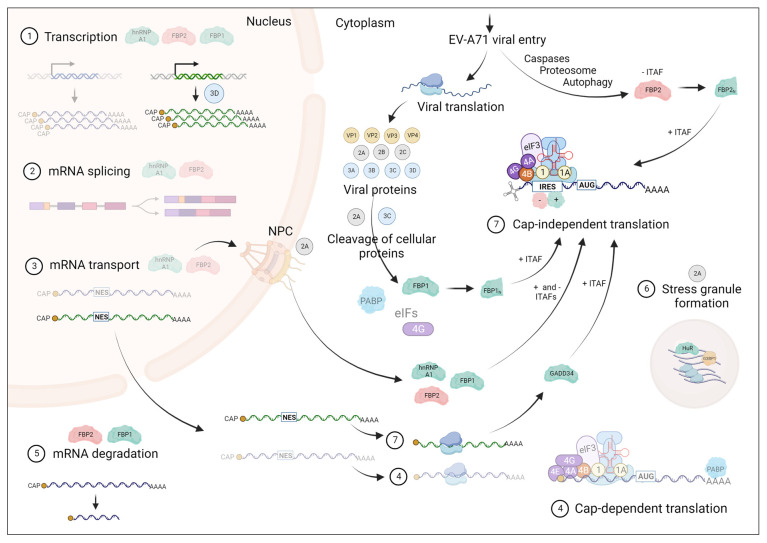
EV-A71 infection perturbs normal RNA metabolism, which leads to a shift in cellular homeostasis. Shown in steps 1–7 are some of the main cellular mechanisms that occur in the host cell. The mechanisms that are illustrated as faded are the ones that are affected by EV-A71 infection and are inhibited specifically due to the relocalization of cellular proteins to be used as ITAFs in EV-A71 IRES-mediated (cap-independent) translation. A few of the ITAFs, hnRNPA1, FBP1, FBP2, GADD34, and HuR, are used to represent the whole repertoire of ITAFs to help explain different viral mechanisms adapted by EV-A71 to facilitate its IRES-mediated translation. Positive ITAFs are colored in green while negative ITAFs are colored in red. These ITAFs are shown at the cellular functions that they regulate prior to EV-A71 infection. Note, a cleaved product of eIF4G is still involved in translation initiation, even though it is shown as faded due to cleavage by viral protease 3C^pro^.

**Table 1 viruses-16-00075-t001:** Cellular proteins (ITAFs) that are repurposed upon EV-A71 infection. The ITAFs, their cellular distributions pre- and post-infection, their regulatory site(s) within the IRES (if known), and their effect on IRES-mediated translation are indicated. The PDB IDs of the proteins (bound to RNA) shown in Figure 3 are indicated in bold.

ITAF	Other Names	Cellular Distribution		Regulatory Site	IRES Activity	PDB ID(s)
		Pre-Infection	Post-Infection			
Heterogeneous nuclear ribonucleoprotein A1	hnRNP A1	Nucleus	Cytoplasm	SLII and IV	Enhancement	**4YOE: 1.92 Å** 5MPL: NMR 5MPG: NMR
AU-rich element RNA-binding protein 1	AUF1, hnRNP D	Nucleus	Cytoplasm	SLII	Inhibition	1WTB: NMR 1 × 0F: NMR
Heterogeneous nuclear ribonucleoprotein K	hnRNP K	Nucleus	Cytoplasm	SLI, II, and IV	-	1J5K: NMR 1ZZI: 1.80Å 1ZZJ: 2.30 Å 7CRE: 3.00 Å
Poly(c)-binding protein 1	PCBP1, hnRNP E1	Nucleus, Cytoplasm	Cytoplasm	SLI and IV	Enhancement	1ZTG: 3 Å 3VKE: 1.77 Å
Polypyrimidine tract-binding protein 1	PTB, PTB1 PTBP1 hnRNP I	Nucleus	Cytoplasm	SLVI + linker (564–742 nt)	Enhancement	**2N3O: NMR** 2AD9: NMR 2ADB: NMR 2ADC: NMR
Far upstream binding protein 2	FBP2, FUBP2 KSRP, KHSRP	Nucleus	Cytoplasm	SLI-SLII (1-167) SLII-SLIII (91-228) SLVI + linker (566–745)	Inhibition	**4B8T: NMR**
C-terminus cleaved FBP2	FBP2_1–503_	-	-	5′-UTR	Enhancement	
N-terminus cleaved FBP2	FBP2_190–711_	-	-	5′-UTR	Inhibition	
Far upstream binding protein 1	FBP1 FUBP1	Nucleus	Cytoplasm	Linker (686–714 nt)	Enhancement	1J4W: NMR
Cleaved FBP1	FBP1_1-371_	-	-	Linker (656–674 nt)	Enhancement	
Argonaute 2	Ago2	Nucleus, Cytoplasm (P-bodies)	-	SLII	Enhancement	**5KI6: 2.15 Å**
Human Antigen R	HuR, ELAVL1	Nucleus, shuttles to Cytoplasm	Cytoplasm	SLII	Enhancement	4ED5: 2.00 Å **6G2K: 2.00 Å** 6GC5: 1.90 Å 6GD2: 1.90 Å
Moloney leukemia virus 10 (C-terminus domain)	MOV10	Cytoplasm	Cytoplasm (P-bodies & aggregates perinuclear)	SLI and IRES (Excluding the linker region)	Enhancement	
Silent mating type information regulation 2 homolog 1	SIRT1	Nucleus	Cytoplasm	SLI, II, III and V	Inhibition	
68-kDa Src-associated protein in mitosis	Sam68, KHDRBS1	Nucleus	Cytoplasm	SLIV and V	Enhancement	
Far upstream element-binding protein 3	FUBP3	Nucleus	Cytoplasm	5′-UTR	Enhancement	
Growth arrest and DNA damage-inducible protein 34	GADD34, PPP1R15A	ER membrane Mitochondial membrane	-	5′-UTR	Enhancement	
DEAD-box protein 3	DDX3	Nucleus Cytoplasm	-	Full 5′-UTR	Enhancement	**6O5F: 2.5 Å**
Apolipoprotein B mRNA-editing enzyme, catalytic polypeptide-like 3G	APOBEC3G, A3G	Cytoplasm (mainly) Nucleus P-bodies	Cytoplasm Virions	SLI and II	Inhibition	5ZVA: 2.30 Å 5ZVB: 2.00 Å 6BUX: 1.86 Å 7UXD: 1.50 Å
Staufen homolog 1	Staufen1	Rough ER Cytoplasm	-	5′-UTR	Enhancement	**6HTU: 2.89 Å**
Heterogeneous nuclear ribonucleoprotein H	HNRNP H	Nucleus	Cytoplasm	5′-UTR	Enhancement	
Heterogeneous nuclear ribonucleoprotein F	HNRNP F	Nucleus	Cytoplasm	5′-UTR	Enhancement	**2KFY: NMR** 2KG0: NMR 2KG1: NMR
Early growth response-1	EGR1	Nucleus	Cytoplasm	SLI and IV	Enhancement	4R2A: 1.59 Å

## Data Availability

Not applicable.

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
