# Peer review of "The Repurposing of Cellular Proteins during Enterovirus A71 Infection"

_viruses, 2023, doi:10.3390/v16010075_

Round 1

Reviewer 1 Report

Comments and Suggestions for Authors

This manuscript presents an overview of the proteins interacting with the IRES element of Enterovirus A71. The authors provide a list of proteins including a brief description of their functions and the positive or negative role in EV-A71 IRES activity. The review is well written although the manuscript would benefit of some clarifications and additions. 

1) In general, proteins are treated individually. The review lacks a connection among the list of factors reported to activate or repress EV-A71 IRES activity. It would be interesting to describe in one section the proteins known to interact simultaneously with the IRES element, stimulating or repressing translation.

2) Related to the previous point, the connection between the activity of the different ITAFs with eIFs is missing.

3) Authors should comment on the specificity of these ITAFs for the EV-A71 IRES. Do they also stimulate or repress other picornavirus IRES and/or other viral IRES elements?

4) Similarly, comments about the high number of ITAFs interacting with SLII are lacking. For instance, these proteins have low RNA-binding specificity, SLII is more accessible, etc?

5) The discussion is not very informative. There is no sufficient discussion about the role of viral proteins and eIFs required for IRES activity. This is a point worth expanding.

Reviewer 2 Report

Comments and Suggestions for Authors

EVA71 is the main causative agent of HFMD, and it has resulted in high morbidity and mortality rates in infants and children under five years old. However, to date, it is no FDA-approved treatment all over the world. Understanding the basic mechanisms of viral processes aids in selecting more efficient drug targets and designing more effective antivirals to thwart this virus. In this review, Sudeshi M. Abedeera et al. comprehensively surveyed the cellular proteins to bind the 5'-UTR and influence viral translation and replication, and emphasized the comparing proteins' functions and localizations pre- and post-EVA71 infection. It is crucial for developing effective antivirals. The manuscript provides a good hint for future researchers. Whereas, some clarifications are still needed to improve the quality of the manuscript and make it more publishable.

1. In the main text, there are many inappropriate abbreviations or lack of full prototypes. What is PDB? If it appears only once in the text, I recommend you don't need to abbreviate, Like WHO, miRNA et al.

2. Line 37-39 Some references failed to provide useful clues. For example, the specific data mentioned could not be found in reference 5.

3. Line 40 Some references do not conform to the normal citation styles. For example, the ordinal position of reference 6.

4. Line 50 There are three inactivated, whole EVA71 vaccines available in China. What about the protective efficacy of the vaccines? I think it should be more clarification here.

5. Some of the viewpoints need to be supported by related references. Like line 75-83 and the whole section of Discussion.

6. Line 149 This paper mainly focused on “The Repurpose of Cellular Proteins during Enterovirus A71 infection”, but some important references don’t involve EVA71 infection. Like Reference 41, it cannot cover the conclusions. So please check the availability of all the references in detail. 

Reviewer 3 Report

Comments and Suggestions for Authors

Overall, this is a really beautiful review of the host factors that affect EV-A71 translation and/or replication.  The figures are really lovely and very helpful to the reader.  This review does a great job at describing the normal function of the host factor, its localization within the cell (before and after infection), and the stage of the viral life cycle that is affected by loss of the host factor. This review is highly informative and very timely.  There will be great interest in such a high-quality review.

General concept comments:

1.     I am curious, when the authors report that a host factor affects translation and replication how are they differentiating an effect on translation that by being upstream of replication enhances replication indirectly.  It seems to this reviewer that most host factors that would enhance IRES activity would result in increased viral proteins and viral replication, but not necessarily affect replication directly.

2.     Figure 4 is not very clear. I am not sure what the mRNA transport from the nucleus to the cytoplasm indicates, then why cap and poly(A) are removed and why is one labeled 4 and the other 7, there is already a 7 in the diagram used for Cap-independent translation.  Perhaps extending the legend to better describe this figure would be helpful. Also, why was viral replication and packaging not included in the figure?

3.     Line 626: What do the authors mean by “to drive viral replication towards predictable outcomes”?

Specific comments:

1.     Line 58 – It might be worth mentioning that type I IRESs are land and scan IRESs. Especially when the reader gets to the section on FBP1 and 2.

2.     Line 59 “The 5´-UTR region is the control hub for EV-A71 genome replication and translation.” Seems like an over simplification of the replication process and could be mis-interpreted. Suggest clarifying or deleting.

3.     Line 61 could be “host” rather than “Host’s”, or change to “the host’s”. Similar changed for line 67.

4.     Line 75-76 “For IRES-mediated translation to take place, 40S ribosomal subunit must be recruited  on to the IRES elements using eIFs and auxiliary RNA-binding proteins (RBPs) which are known as IRES trans-acting factors (ITAFs).” I did not know that ITAFs have been shown to recruit ribosomes, if so these references need to be added. More likely ITAFs help stabilize IRES structure for 40S recruitment, but I am unaware of any studies that show an interaction between ITAFs and ribosomes.

5.     Line 77: more accurately, most ITAFs are nuclear or cycle between the nucleus and cytoplasm.

6.     Figure 1 legend and lines 353 & 564. The authors may consider replacing transcription with replication since enteroviruses do not have any sub-genomic RNAs that are transcribed for translation rather – Strand and + Strand synthesis is part of the replication phase.

7.     Figure 2 and 3 legends.  Nucleotides should not be capitalized.  Also, for clarity the authors might consider stating that base paired nucleotides are gray.  Also, it would be easier for the reader if the labels were re-phrased as “adenosine (red), uracil (green), guanine (blue), cytosine (yellow)”.

8.     Figure 3D and line 127:  The authors report that the hnRNPA1 binds to a YAG sequence however in figure 3D the RNA is green blue red – which would be a YGA, or an AGY according to their color scheme. If it bound to a YAG then it should be green, red, blue or blue, red, green. It is noted than in Figure 2 there is a green, red, blue single stranded region, but not a green blue red one, suggesting Figure 3D may need to be corrected.

9.     Figure 3: the authors may consider either labeling the images with the protein and SL region or adding the SL region to the legend.

10.  Lines 206-207 says that PTB interacts with CA rich regions and references Figure 3e which shoes a green and yellow RNA, which would be a UC rich region according to the legend.

11.  Lines 181, 200, 293, 304-306, authors should state how much viral production was reduced.  

12.  Line 406: explain what PEST repeats are.

13.  Line 580 could the author clarify which steps of viral replication occur in the nucleus or what is meant by using the word “most” in that sentence.

14.  It would be useful to include, here, how much the drug, DMA-135, reduces viral titers.

15.  Line 212 delete “have”, 213 IRESes, line 235 the bold formatting is odd as it makes it seem like this is a major heading when it is a sub-heading under 2.6. line 498-90 should be hnRNP H1 and hnRNP F, line 540 add “and”, also capitalization of protein names in that paragraph is not necessary.

Comments on the Quality of English Language

I found at times articles "the" or "a" would have improved readability.

Verb use and tenses could be improved.  Generally published work is usually in the present tense as in, “ITAFs bind to the 5’UTR” while unpublished work is generally past tense. Since this is a review of the literature most verbs should be present tense for example line 305 could be changed to:HuR knockdown reduces 3Cpro expression and viral yields”. This would also help with reading and flow.

Remove contractions
